# $K^*$ and Partial Order Reduction for Top-Quality Planning

## Michael Katz, Junkyu Lee

IBM T.J. Watson Research Center
1101 Kitchawan Rd, Yorktown Heights, NY 10598, USA
{michael.katz1, junkyu.lee}@ibm.com

## Abstract

Partial order reduction techniques are successfully used for various settings in planning, such as classical planning with $A^*$ search or with decoupled search, fully-observable non-deterministic planning with $LAO^*$, planning with resources, or even goal recognition design. Here, we continue this trend and show that partial order reduction can be used for top-quality planning with $K^*$ search. We discuss the possible pitfalls of using stubborn sets for top-quality planning and the guarantees provided. We perform an empirical evaluation that shows the proposed approach to significantly improve over the current state of the art in unordered top-quality planning. The code is available at https://github.com/IBM/kstar.

## Introduction

The need for producing a set of top-quality plans is well established by many real-life applications, including plan recognition (Sohrabi, Riabov, and Udrea 2016), malware detection (Boddy et al. 2005), business process automation (Chakraborti et al. 2020), and automated machine learning (Katz et al. 2020). Top-quality planning supplies the demand, generating all high-quality plans up to a certain bound (Katz, Sohrabi, and Udrea 2020). If some plans are considered equivalent from an application perspective, as is often the case with plans that are equivalent if operator orderings are ignored, it can be sufficient to cover the equivalence classes instead of all plans (Katz, Sohrabi, and Udrea 2020; Katz and Sohrabi 2022). This so-called unordered top-quality planning serves, among other, as a basis for solving additional computational problems, such as quality-aware diverse planning (Nguyen et al. 2012; Vadlamudi and Kambhampati 2016; Katz, Sohrabi, and Udrea 2022). While this is clearly an important computational problem of high practical value, the selection of available planners is quite limited. The two existing approaches are based on Forbid Iterative (Katz et al. 2018), an approach that iteratively solves a modified planning problem with classical planners, decreasing the set of solutions in each iteration, and SymK (Speck, Mattmüller, and Nebel 2020), a symbolic search based approach. While in top-k planning, a $K^*$ search-based approach (Aljazzar and Leue 2011) was recently shown to pro-

duce competitive results (Lee, Katz, and Sohrabi 2023), it was not applied to unordered top-quality planning.

In this work, we propose a competitive unordered top-quality planner based on $K^*$ search and partial order reduction. Partial order reduction was successful for various settings in planning, such as classical planning with $A^*$ search or with decoupled search (Gnad, Hoffmann, and Wehrle 2019), fully-observable non-deterministic planning with $LAO^*$ (Winterer et al. 2017), planning with resources (Wilhelm, Steinmetz, and Hoffmann 2018), or even goal recognition design (Keren, Gal, and Karpas 2018). They were never, however, applied in the setting of unordered top-quality planning, arguably among the more naturally fitting problem, where different orderings of the same plan are of no added value. Focusing on stubborn sets (Valmari 1989), we show that $K^*$ search over the space reduced with strong stubborn sets (Wehrle and Helmert 2012; Alkhazraji et al. 2012; Wehrle and Helmert 2014) is safe to use for unordered top-quality planning. We show that no adaptation to these techniques is needed if a single-goal planning task transformation is used. Such transformations are required for $K^*$ search and must be performed anyway. We point out the weaknesses of the transformation used in previous implementations of $K^*$ (Katz et al. 2018) and suggest using a recently proposed transformation for symmetry pruning variant (Katz and Lee 2023), which preserves the pruning power of the original planning task. We perform an experimental evaluation that shows a significant increase in performance over the current state of the art and establishes our approach as the new state of the art for unordered top-quality planning.

## Background

We introduce the necessary concepts in top-quality planning, $K^*$ search, as well as partial order reduction.

### Top-quality Planning

We consider classical planning tasks in the well-known SAS$^+$ formalism (Bäckström and Nebel 1995), extended with action costs. Such *planning tasks* $\Pi = \langle \mathcal{V}, \mathcal{O}, s_0, s_\star \rangle$ consist of a finite set of finite-domain *state variables* $\mathcal{V}$, a finite set of *actions* $\mathcal{O}$, an *initial state* $s_0$, and the *goal* $s_\star$. Each variable $v \in \mathcal{V}$ is associated with a finite domain $dom(v)$ of variable values. A *partial assignment* $p$ maps a subset of

variables $vars(p) \subseteq \mathcal{V}$ to values in their domains. For a variable $v \in \mathcal{V}$ and partial assignment $p$, the value of $v$ in $p$ is denoted by $p[v]$ if $v \in vars(p)$ and we say $p[v]$ is *undefined* if $v \notin vars(p)$. A partial assignment $s$ with $vars(s) = \mathcal{V}$, is called a *state*. State $s$ is *consistent* with partial assignment $p$ if they agree on all variables in $vars(p)$, denoted by $p \subseteq s$. $s_0$ is a state and $s_\star$ is a partial assignment. A state $s$ is called a *goal state* if $s_\star \subseteq s$ and the set of all goal states is denoted by $\mathcal{S}_{s_\star}$. Each action $o$ in $\mathcal{O}$ is a pair $\langle pre(o), eff(o) \rangle$ where $pre(o)$ and $eff(o)$ are partial assignments called *precondition* and *effect*, respectively. Further, $o$ has an associated *cost* $C(o) \in \mathbb{R}^{0+}$. An action $o$ is applicable in state $s$ if $pre(o) \subseteq s$. Applicable in $s$ actions are denoted by $\mathcal{O}(s)$. Applying $o$ in $s$ results in a state denoted by $s[\![o]\!]$ where $s[\![o]\!][v] = eff(o)[v]$ for all $v \in vars(eff)$ and $= s[\![o]\!][v] = s[v]$ for all other variables. An action sequence $\pi = \langle o_1, \cdots, o_n \rangle$ is applicable in state $s$ if there are states $s_0, \cdots, s_n$ such that $o_i$ is applicable in $s_{i-1}$ and $s_{i-1}[\![o_i]\!] = s_i$ for $0 \leq i \leq n$. We denote $s_n$ by $s[\![\pi]\!]$. An action sequence with $s_0[\![\pi]\!] \in \mathcal{S}_{s_\star}$ is called a *plan*. The cost of a plan $\pi$, denoted by $C(\pi)$ is the summed cost of the actions in the plan. The set of all plans is denoted by $\mathcal{P}_\Pi$. A plan is *optimal* if its cost is minimal among all plans in $\mathcal{P}_\Pi$. Cost-optimal planning deals with finding an optimal plan or proving that no plan exists (the task is *unsolvable*).

Extending cost-optimal planning, top-quality planning (Katz, Sohrabi, and Udrea 2020) deals with finding *all* plans of up to a specified cost. Formally, the **top-quality** planning problem is as follows. Given a planning task $\Pi$ and a number $q \in \mathbb{R}^{0+}$, find the set of plans $P = \{\pi \in \mathcal{P}_\Pi \mid cost(\pi) \leq q\}$. In some cases, an equivalence between plans can be specified, allowing to possibly skip some plans, if equivalent plans are found. The corresponding problem is called **quotient top-quality** planning and it is formally specified as follows. Given a planning task $\Pi$, an equivalence relation $N$ over its set of plans $\mathcal{P}_\Pi$, and a number $q \in \mathbb{R}^{0+}$, find a set of plans $P \subseteq \mathcal{P}_\Pi$ such that $\bigcup_{\pi \in P} N[\pi]$ is the solution to the top-quality planning problem. The most common case of such an equivalence relation is when the order of actions in a valid plan is not significant from the application perspective. In other words, when you can re-order some of the actions in a plan and still get a valid plan. The corresponding problem is called **unordered top-quality** planning and is formally specified as follows. Given a planning task $\Pi$ and a number $q \in \mathbb{R}^{0+}$, find a set of plans $P \subseteq \mathcal{P}_\Pi$ such that $P$ is a solution to the quotient top-quality planning problem under the equivalence relation $U_\Pi = \{(\pi, \pi') \mid \pi, \pi' \in \mathcal{P}_\Pi, \text{MS}(\pi) = \text{MS}(\pi')\}$, where $\text{MS}(\pi)$ is the multi-set of the actions in $\pi$

## $K^*$ Search for Top-quality Planning

Given a top-quality planning problem $\langle \Pi, q \rangle$, $K^*$ first performs $A^*$ until the cost bound is reached, and then applies Eppstein's Algorithm ($EA$) to the search graph revealed by $A^*$. We omit here the details of $EA$ as they are not necessary for this paper and refer a curious reader to Aljazzar and Leue (2011) and Eppstein (1998) for details. One important property that we emphasize is that $EA$ can enumerate all paths in the search graph in the order of their costs.

One of the limitations of $K^*$ is its restriction to graphs with a single goal state. In planning, however, tasks can have many goal states. In cases when the partial assignment $s_\star$ is not a full state, it is possible to transform $\Pi$ into a planning task with a single goal state. One way to achieve that is to add one binary variable $v_g$ to indicate whether a goal was reached. Further, we add one zero-cost action $o_g$ with precondition $pre(o_g) = s_\star \cup \{v_g = 0\}$ and effect $eff(o_g) = \{v_i = t[v_i] \mid v_i \in vars(t)\} \cup \{v_g = 1\}$ for an arbitrary full state $t$ over the original variables. Additionally, each original action precondition, as well as the initial state are extended with $\{v_g = 0\}$. Finally, the goal is set to $eff(o_g)$, making it a full variable assignment. For a planning task $\Pi$, we denote its single goal transformation by $\Pi_g$. In words, the additional goal-achieving zero-cost action can be applied once (and only once) the original goal was achieved, changing the state to the new goal state. No action is applicable in the new goal state, and therefore there is one-to-one correspondence between the plans of $\Pi$ and those of $\Pi_g$. The transformation was used, albeit not described by Katz et al. (2018), in their implementation of $K^*$. It is worth noting that domain independent heuristics for planning can be quite sensitive to such transformations.

## Partial Order Reduction

A central to partial order reduction techniques is the notion of *safe* successor pruning (Wehrle and Helmert 2014). A successor pruning function $succ$ for a planning task $\Pi$ is *safe*, if for every state $s$, the cost of an optimal solution for $s$ is the same when using the pruned state space induced by $succ$ as when using the full state space. When using safe successor pruning, it is possible to search the pruned state space instead when searching for cost-optimal plans. Stubborn sets (Wehrle and Helmert 2012; Alkhazraji et al. 2012) induce safe successor pruning functions by helping identifying actions that can safely be ignored at node expansion. It is done by specifying a set, such that if an applicable action is not in the set, it can be safely ignored (e.g., Wehrle and Helmert 2014).

At the core of these partial order reduction techniques is the idea that, for each non-goal state $s$, if a goal is reachable from $s$, then at least one *strongly optimal* (an optimal plan with a minimal number of 0-cost actions among all optimal plans) is preserved in the pruned state space.

Two main notions in stubborn sets are *interference* and *necessary enabling sets* (NES). Interference dictates whether two actions disable each other or conflict. Necessary enabling sets for an action $o$ and a set of paths from the initial state is a set of actions that appear on the paths that include $o$ before its first appearance.

**Definition 1 (GSSS)** *Let $\Pi$ be a planning task and $s$ be a solvable non-goal state. Let $\overline{S}$ be the states along strongly optimal plans for $s$. A set $S \subseteq \mathcal{O}$ is a GSSS for $s$ if:*

*(i) $S$ contains at least one action from at least one strongly optimal plan for $s$.*

*(ii) For every $o \in S \setminus \mathcal{O}(s)$, $S$ contains a NES for $o$.*

*(iii) For every $o \in S \cap \mathcal{O}(s)$, $S$ contains all $o' \in \mathcal{O}$ that interfere with $a$ in any state $s \in \overline{S}$.*

## Stubborn Sets for Top-quality Planning

In this section, we show that $K^*$ search can be used for solving the unordered top-quality planning problem. To do that, we need to ensure that the stubborn sets are well-defined for all states expanded by the search algorithm. For non-goal states, this is true. For goal states, however, stubborn sets may prune all plans. Therefore, it is crucial that the search algorithm does not expand goal states. For $K^*$ this is indeed the case, since it solves the single-goal transformation $\Pi_g$ of the input planning task $\Pi$, and in the single goal transformation of Katz et al. (2018), the (only) goal state does not have successors, so plans do not traverse through the goal state. Therefore, from now on, we focus on non-goal states only.

We start with an observation that existing work (e.g., Alkhazraji et al. 2012) proves a stronger property than preserving at least one optimal plan. Let us state the property explicitly. Let $\Pi$ be a planning task, $s$ be some state and $\pi_s = o_1, \ldots, o_n$ be some plan for $s$. Let $f : \mathcal{S} \mapsto P(\mathcal{O})$ be a strong stubborn set and $\mathcal{O}_f(s) := \mathcal{O}(s) \cap f(s)$ be the corresponding subset of applicable actions. If $o_1 \notin \mathcal{O}_f(s)$, let $i$ be the smallest index such that $o_i \in \mathcal{O}_f(s)$. Then, $\pi'_s = o_i, o_1, \ldots, o_{i-1}, o_{i+1}, \ldots, o_n$ obtained from $\pi_s$ by moving the action $o_i$ to the front, is also a plan for $s$.

It is worth noting that this property is also strongly related to the *operator shifting property* (Sievers and Wehrle 2021).

We use this observation to prove the following theorem.

**Theorem 1** *Let $\mathcal{T}$ be the transition system of $\Pi_g$, $f : \mathcal{S} \mapsto P(\mathcal{O})$ be a strong stubborn set, and $\mathcal{T}_f$ be the corresponding reduced transition system. For each plan $\pi$ of $\Pi_g$, there exists a reordering $\pi'$ that corresponds to a goal path in $\mathcal{T}_f$.*

**Proof:** We construct a plan $\pi'$ from $\pi = o_1, \ldots, o_n$ iteratively applying the observation above. We start with $\pi'$ being an empty sequence. Let $s$ be the current state, starting with the initial state $s_0$ and let $\pi_s$ be a plan for the current state, starting with $\pi$. While $\pi_s$ is not an empty sequence, we find the next action to add to $\pi'$ as follows. Let $o$ be the first action from $\pi_s$. If $o$ is in $\mathcal{O}_f(s)$, then we add $o$ to $\pi'$ and update the current state to be $s[\![o]\!]$. We remove the first action from $\pi_s$, maintaining $\pi_s$ to be a plan for $s$. If $o$ is not in $\mathcal{O}_f(s)$, we use the observation to find $o_i \in \mathcal{O}_f(s)$ and a plan $\pi'_s$ for $s$ that is a reordering of $\pi_s$ that starts with $o_i$. We then add $o_i$ to $\pi'$ and update the current state to be $s[\![o_i]\!]$. We replace $\pi_s$ with $\pi'_s$ and then remove the first action from $\pi_s$, maintaining $\pi_s$ to be a plan for $s$.

Every iteration, the length of $\pi_s$ is reduced by 1 action, moving the action to $\pi'$. Thus, the concatenation of $\pi'$ and $\pi_s$ at each step is a plan, a reordering of the plan $\pi$.  $\square$

A corollary from Theorem 1 is that we can solve unordered top-quality planning by finding plans in the reduced transformed transition system, using strong stubborn sets for pruning the search.

Next, we question whether the stubborn sets applied to the single-goal transformation have the same pruning power as if they were applied to the original task. Looking closer at the transformation, since the goal state may include variable values that are achieved by original actions, these actions

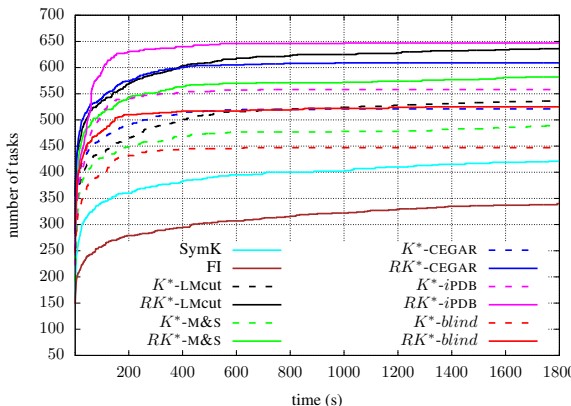

Figure 1: Anytime performance of tested configurations.

(and therefore some other) may be added to the stubborn set, making it unnecessarily large. The issue can be avoided if a different transformation is used. Recently, such a transformation was proposed for a similar purpose, to preserve the pruning power of symmetries (Katz and Lee 2023). Their transformation differs in the goal state (and consequently the effect of the goal achieving action), defining new goal values for all original variables. With that, the issue mentioned above disappears. The issue still persists in the states of $\Pi_g$ that correspond to goal states of $\Pi$. There, the action $o_g$ will be added to the stubborn set, and so will all other actions of $\Pi_g$, since they interfere with $o_g$. For any other non-goal state $s$ of the transformed problem $\Pi_g$, such that a goal is reachable from $s$, let $s'$ be the corresponding (non-goal) state of the original problem $\Pi$. Strong stubborn set algorithms applied to $\Pi_g$ will produce the same *applicable* action set for $s$ as the set obtained for $s'$ when applied to $\Pi$. To see that, observe that $o_g$ is the only action that achieves the goal in $\Pi_g$. It interferes with all other actions of $\Pi_g$ and its precondition, except for on $v_g$ agrees precisely with the goal of $\Pi$. All original actions interact with each other in $\Pi_g$ in precisely the same way as in $\Pi$. Once any applicable action is added to the stubborn set of $s$, so will $o_g$ (interference). Since $o_g$ is not applicable, this will trigger adding the *necessary enabling set* for $o_g$. Assuming that the choice of finding an unsatisfied goal and of finding an unsatisfied precondition of $o_g$ are resolved the same way, we get the same behavior.

## Experimental Evaluation

We have integrated the existing implementation of partial order reduction techniques into an existing implementation of $K^*$ algorithm (Lee, Katz, and Sohrabi 2023), built on top of the Fast Downward planning system (Helmert 2006). All experiments were performed on Intel(R) Xeon(R) Gold 6248 CPU @ 2.50GHz machines, with the timeout of 30 minutes and memory limit of 8GB per run. The benchmark set consists of all benchmarks from optimal tracks of International Planning Competitions 1998-2018, a total of 1827 tasks in 65 domains. We have experimented with four admissible heuristics, LMcut (Helmert and Domshlak 2009), merge-and-shrink abstraction (denoted by M&S)

| | | SymK | FI | LMcut | | M&S | | CEGAR | | $i$PDB | | blind | |
|---|---|---|---|---|---|---|---|---|---|---|---|---|---|
| | | | | $K^*$ | $RK^*$ | $K^*$ | $RK^*$ | $K^*$ | $RK^*$ | $K^*$ | $RK^*$ | $K^*$ | $RK^*$ |
| | SymK | 0 | **31** | 5 | 5 | 7 | 7 | 6 | 6 | 3 | 3 | 17 | 16 |
| | FI | 12 | 0 | 6 | 2 | 7 | 3 | 7 | 3 | 7 | 3 | 7 | 3 |
| LMcut | $K^*$ | **39** | **43** | 0 | 1 | **14** | 13 | **12** | 13 | 6 | 6 | **31** | 29 |
| | $RK^*$ | **41** | **48** | 12 | 0 | **25** | 18 | **23** | 18 | 18 | 11 | **39** | 34 |
| M&S | $K^*$ | **35** | **41** | 7 | 6 | 0 | 1 | 11 | 10 | 0 | 0 | **27** | 25 |
| | $RK^*$ | **37** | **45** | 14 | 6 | 10 | 0 | 19 | 14 | 9 | 4 | **34** | 30 |
| CEGAR | $K^*$ | **32** | **40** | 4 | 4 | 12 | 12 | 0 | 1 | 5 | 5 | **27** | 25 |
| | $RK^*$ | **34** | **45** | 14 | 4 | 22 | 14 | 10 | 0 | 15 | 6 | **34** | 29 |
| $i$PDB | $K^*$ | **44** | **43** | 17 | 15 | 19 | 19 | 19 | 19 | 0 | 0 | **35** | 33 |
| | $RK^*$ | **46** | **47** | 26 | 16 | 29 | 21 | 28 | 21 | 10 | 0 | **42** | 37 |
| blind | $K^*$ | 23 | **38** | 2 | 1 | 6 | 6 | 2 | 1 | 1 | 1 | 0 | 0 |
| | $RK^*$ | 25 | **42** | 8 | 2 | 13 | 7 | 8 | 2 | 8 | 1 | **10** | 0 |
| Overall Coverage | | 421 | 341 | 536 | 636 | 489 | 582 | 521 | 609 | 558 | **647** | 447 | 525 |

Table 1: Pair-wise domain level comparison of (unordered) top-quality planners. Each entry in the table represents the number of domains where the row configuration achieves better coverage than the column one. The last row depicts the overall coverage.

(Helmert, Haslum, and Hoffmann 2007), counterexample-guided Cartesian abstraction refinement (denoted by CE-GAR) (Seipp and Helmert 2018), and pattern database heuristic iPDB (Haslum et al. 2007). As planning heuristics can be sensitive to task formulation, we follow the suggestion of Lee, Katz, and Sohrabi (2023) and evaluate the heuristics on the original task. The found plans are added to the solution, checking duplicates as de-ordered multi-sets. Vanilla $K^*$ is compared to $K^*$, pruned by atom-centric stubborn sets (Röger et al. 2020), denoted by $RK^*$. We also compare to the existing unordered top-quality planners, ForbidIterative (FI) (Katz et al. 2018) and SymK (Speck, Mattmüller, and Nebel 2020). Importantly, all planners perform the same translation from PDDL to $SAS^+$, solving the same $SAS^+$ planning problem. In our experiments, we disabled the writing of plans to disk, excluding the write time from our evaluation. All experiments are using the cost multiplier $q = 1$: find all cost-optimal plans, modulo reorderings.

We measure coverage, giving the score of 1 to each task if the unordered top-quality solution was found withing the 30 minutes time bound. Additionally, we measure the time until the solution was found. Figure 1 depicts the any-time overall coverage performance of the tested planners. The lines for the same heuristic are depicted with the same color, solid for $RK^*$ and dashed for $K^*$. For very small timeouts, up to 4s, the best performing overall configuration is $RK^*$ with the LMcut heuristic. From 4s to 60s, $RK^*$ with CEGAR takes the lead. Starting from 60s an onwards, $RK^*$ with iPDB becomes the best performer. It is worth noting that it reaches its almost maximal coverage of 646 out of 647 tasks already at 540s and the maximal coverage by 995s.

Switching now to domain-level coverage, Table 1 shows the pairwise comparison of the tested approaches. Each entry denotes the number of domains where the row planner achieves a better summed coverage than the column planner. Additionally, the last row denotes the overall coverage for each planner. The winners are marked in bold. For the pairwise comparison, the value in (x,y) is bolded if it is larger than the value in (y,x), that is planner x excelled over planner y in more domains than planner y excelled over planner x. Observe that $RK^*$ consistently outperforms $K^*$, with the difference in the number of domains with superior performance being between 9 and 11. $RK^*$ with iPDB is the absolute per-domain winner, as well as overall winner, achieving the overall coverage of 647.

At per-domain level, as expected, $RK^*$ with iPDB does not absolutely dominate other approaches. There are 17 domains where the configuration does not achieve maximal coverage, including AIRPORT and LOGISTICS-IPC2, where FI excels, and BLOCKSWORLD and VISIT-ALL where SymK achieves the largest coverage. Other heuristics also exhibit superior performance: LMcut in DRIVERLOG, ORGANIC-SYNTHESIS-SPLIT, PATHWAYS-NN, PIPESWORLD-NOTANKAGE, SATELLITE, TRUCKS, and WOODWORKING, M&S in PSR-SMALL and WOOD-WORKING, and CEGAR in FREECELL, MPRIME, MYS-TERY, PIPESWORLD-NOTANKAGE, and TRUCKS. Interestingly, the highest coverage in ORGANIC-SYNTHESIS-SPLIT is achieved by $K^*$ with LMcut, while $RK^*$ loses coverage due to the overhead of the partial order reduction used.

## Conclusions and Future Work

In this work, we exploit partial order reduction techniques to improve the performance of $K^*$ search. For that, we show that no adaptation to these techniques is required if a single-goal planning task transformation is used. We point out the weaknesses of the transformation used in previous implementations of $K^*$ and suggest using a recently proposed for the purpose of symmetry pruning variant, which preserves the pruning power of the original planning task. We perform an experimental evaluation that establishes our approach as the new state of the art for unordered top-quality planning.

In the future work, we intend to improve top-k planning with partial order reduction. To do so, an efficient reconstruction of pruned reorderings must be derived. A previously suggested naive exhaustive DFS search (Katz et al. 2018) was found to be too slow to be competitive. Efficient methods can exploit the information on which actions were actually pruned at each state.

## Acknowledgements

We express our gratitude to Malte Helmert for his invaluable contributions through multiple discussions, which greatly contributed to shaping this paper.

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
