# OpenReview forum: "K* and Partial Order Reduction for Top-quality Planning"
_icaps-conference.org/ICAPS/2023/Workshop/HSDIP — ICAPS HSDIP 2023_

### Official Review · Reviewer_vCDc · 2023-04-25
**Good paper, some formal details are missing**

**Rating:** 8
**Confidence:** 4

**Review:**

I previously reviewed a version of this paper for SOCS and copy the relevant parts from that review:

The paper introduces partial-order reduction via strong stubborn sets to top-quality planning using the K* algorithm.
In top-quality planning, the goal is to find all solutions up to a given cost bound. In particular, the paper tackles
the *unordered* top-quality planning problem, where plans that are equal up to re-ordering of actions are considered equivalent.
Partial-order reduction methods such as strong stubborn sets are used in optimal planning to prune the search space admissibly.
They are well-suited for unordered top-quality planning, as they do so by pruning actions that start different permutations
of the same plan in a given search state.

The paper gives the background on planning, top-quality planning, K*, and partial-order reduction.
Then, the approach is described, proven correct, and evaluated empirically.

The main issue that I see with this work is the lack of formality when it comes to describing the approach and proving it correct.
This starts with the lack of a definition of what a stubborn set even is. This is only described informally.
Important concepts such as the transition system of a planning task, and the pruned transition system under stubborn sets are not defined, either.
For a paper that introduces stubborn-set pruning for K* search, I would have expected a proper definition of what a stubborn set is.
In my opinion, too much emphasis is put on the experiments. Given that the paper does not make use of the 8-page limit for HSDIP, a proper formalization of the underlying base methods could have been added.

A second shortcoming is that the paper does not discuss the effectiveness of the stubborn-set pruning. It is clear
that the search space is reduced, but it is not clear to which extent. This is especially interesting since
stubborn sets are not a perfect method to reduce partial orders, i.e. there are plan reorderings that it cannot capture.
I think this should be mentioned when introducing the approach, and ideally be evaluated, too.

The paper is a nice fit for the workshop. The evaluation is very convincing, but the approach could be formalized more thoroughly.
Hence, I clearly recommend acceptance.

---

### Official Review · Reviewer_rh5Y · 2023-04-26
**Interesting approach, light on details**

**Rating:** 7
**Confidence:** 4

**Review:**

Summary:

This paper presents the use of stubborn sets to reduce state spaces in order to do top-quality planning. The K* algorithm is used, which employs A* to search until a cost bound is reached, and then applies Eppstein's Algorithm to discover additional plans. The work uses stubborn sets to prune successors safely in order to still find solutions for unordered top quality planning. They demonstrate that K* in this reduced state space can achieve better results than existing state of the art algorithms across many planning domains.


Specific Feedback:

This work demonstrates a useful technique for an interesting problem in planning, but is light on much of the background and details. There are several concepts that would benefit from at least a summary, such as stubborn sets and Eppstein's Algorithm. In particular, it would be helpful to clarify why Eppstein's Algorithm is beneficial after the A* search in this setting.

The results clearly show improvement in the performance of K* when using stubborn set pruning, using a variety of heuristics. The paper would benefit from some further analysis of when stubborn set pruning is most likely to be beneficial. Is there something particular about the problem domains in which RK* gets better results than K*? There is attention given to which domains RK* with iPDB gets poorer results than other configurations, but giving insight into when the reduction will improve performance seems more salient.

Overall, the paper presents a new, interesting, and successful approach to top quality planning and is worth accepting to the workshop. However, I would encourage the authors to expand on the background and definitions to make the paper more clear, and if possible to give more explanation for when this reduction is likely to be of use.


Minor points:

Where it is stated "we omit here the details of EA", perhaps a sentence or two summary would be beneficial.

Under Partial Order Reduction, "A central to" seems to be missing a word. "A central concept to..." perhaps?

Also under Partial Order Reduction, "at least one strongly optimal" appears to be missing the word "plan" either before or after the parenthetical.

Before or at the beginning of the section on stubborn sets, a short summary of stubborn sets would improve clarity.

---

### Decision · Program_Chairs · 2023-05-06

**Decision:**

Accept

**Comment:**

We are happy to announce that the paper is accepted to the workshop. Both reviewers clearly stated that the work is very relevant for the workshop.

For the final version, we ask that you address the comments made by the reviewers. We especially ask that you make use of the space available to include more details on the topics raised by the reviewers.